# Watershed-Mediated Ecomorphological Variation: A Case Study with the Twin-Striped Clubtail Dragonfly (*Hylogomphus geminatus*)

**DOI:** 10.3390/insects14090754

**Published:** 2023-09-09

**Authors:** Joseph S. Girgente, Nancy E. McIntyre

**Affiliations:** Department of Biological Sciences, Texas Tech University, Lubbock, TX 79409, USA

**Keywords:** aquatic insects, environmental changes, body length, landscape, land-cover, streams

## Abstract

**Simple Summary:**

Changes in the size of aquatic or amphibious organisms can signal environmental alterations. The Twin-striped Clubtail is a dragonfly in the southeastern US with known variation in total body length. Our objective was to characterize the significance and extent of size differences in the species and determine whether those differences are associated with watersheds across the species’ limited range. In this species, total body length varied by watershed. Significant differences in length were noted in many watershed comparisons, most notably in adjacent watersheds on either side of the Apalachicola River, FL, which differed in land-cover types; smaller individuals were associated with more disturbed forms of land cover. Our results suggest there are watershed-level differences in body length across the range of the species that should be examined more closely with respect to different potential environmental stressors, such as poor water quality from changing land cover associated with agriculture and urbanization.

**Abstract:**

Anthropogenic land-cover change is modifying ecosystems at an accelerating rate. Changes to ecomorphologically variable taxa within those ecosystems serve as early-warning signs that resources on which humans and other animals depend are being altered. One known ecomorphologically variable taxon is *Hylogomphus geminatus*, a species of dragonfly in the southeastern United States that shows pronounced variation in total body length across its limited geographic range. We measured total length of live as well as preserved museum specimens of *H. geminatus* and the sympatric species *Progomphus obscurus* (as a means for comparison). Both species showed significant size differences linked to HUC-8 watersheds in which they occur. *H. geminatus* showed additional significant differences on either side of the Apalachicola River, Florida, for all comparisons by sex. In overlapping watersheds, the species tended to show the same trends in length relative to their respective averages. Smaller body length was associated with more urban and agricultural land cover. These findings indicate that ecomorphological variation is tied to the watershed scale and point to significant variations on either side of the Apalachicola River. More thorough future analyses would be needed to verify trends in body length and identify the drivers behind them.

## 1. Introduction

Because species’ physical traits are shaped by environmental conditions, trait variation can inform extinction risk [1,2]. Thus, examining patterns of ecomorphological variation may improve our ability to predict how species may respond to environmental changes, such as those stemming from anthropogenic land-cover changes. Land-cover changes have altered water quality and availability in many parts of the world [3], which may be reflected in development and subsequent body morphology of some animals [4]. Identifying environmental factors influencing ecomorphologically variable taxa is thus crucial in understanding effects of anthropogenic activities and landscape disturbance on aquatic resources. 

Ectotherms such as aquatic insects are especially sensitive bioindicators that respond to anthropogenic impacts on both abiotic and biotic environmental variables [5,6]; members of the insect order Odonata (dragonflies and damselflies) have been noted as particularly useful gauges of environmental conditions [7]. With relatively short generation times, their morphological traits (e.g., body size) may signal recent environmental changes [8] but environmental factors that affect ecomorphology in odonates in the field are poorly understood. Some abiotic variables that influence insect morphology include aquatic factors like temperature and dissolved oxygen (DO) [9,10,11]; other variables such as depth, conductivity, pH, or turbidity may also affect development and, thus, body size [12,13,14]. Moreover, some biotic variables, like prey availability, are known to have relationships with size [15]. As odonates spend much of their lives in water, these aquatic variables should have a larger impact on their morphology than would terrestrial variables because odonates do not continue growing once they have emerged from the water as winged adults. The length of time spent as an aquatic nymph is variable and depends on species-specific life history, climatic factors such as temperature, water persistence, availability of food, and other factors, but can range from roughly 3 weeks to over 3 years [16]. Once an adult, dispersal allows gene flow across populations, so consistent ecomorphological patterns in populations indicate the presence of environmental drivers of variation. 

Some of these drivers of variation may be due to land-cover changes, which are occurring at an accelerating rate in many parts of the world. Urban development, deforestation and waterway modifications change aquatic habitats and induce stressors like increasing pollutants, temperatures, or stream velocity, or they may reduce habitat entirely, leaving large swathes of unsuitable land cover that act as barriers to even vagile dragonfly species. Landscape changes associated with environmental consequences and modifications of hydrology could be responsible for variation in odonate morphology through land-cover changes [17], including urban development, deforestation, streambank destabilization, and stream channelization.

One known ecomorphologically variable odonate is *Hylogomphus geminatus* (Carle, 1979) the Twin-striped Clubtail (Gomphidae), a semivoltine species [18] (p. 219) that primarily inhabits relatively small, sandy-bottomed streams across the panhandle of Florida and surrounding areas along the Gulf Coast of the United States. In Carle’s original description of *H. geminatus* [19], he noted differences in body length in adults from populations on either side of the Apalachicola River, Florida, but did not suggest any mechanisms for these differences [19]. Subsequent unpublished observations have noted these differences in *H. geminatus* as well [20]. Although the river spans several hundred feet across much of its width [21], that distance should not act as a barrier to gene flow for vagile organisms such as odonates. 

Some intraspecific body size variation in odonates (with size encapsulating aspects of total body length, length of specific anatomical features, and mass) is the result of nymphal development time, with longer time spent as a nymph generally producing larger adults [22], although nymphs may actually shorten their development time under certain environmental conditions to avoid emerging at a smaller size [23]. The duration of nymphal development time is strongly driven by environmental conditions [24,25,26,27]. It is well-documented in adult odonates that body size is typically positively associated with thermoregulatory ability, competitive ability, and fecundity [28,29,30,31]. Thus, the implications of differences in length in *H. geminatus* across populations that are being driven by environmental conditions include potential mate limitation (if different populations are emerging at different times because of different nymphal development times) and access to resources. The major river drainage within the core range of *H. geminatus* is the Apalachicola River, a crucial freshwater resource for the surrounding biotic community, with a drainage that includes four major rivers spanning three U.S. states and an area of almost 51,800 sq km [32]. Anecdotally, it has been observed that smaller individuals west of the Apalachicola River tended to emerge earlier in the year than the larger individuals east of the river [20]. 

Although some dragonflies are capable of crossing oceans during annual migratory events, these species tend to occupy lentic habitats whereas many members of the gomphid family occupy lotic habitats and are not known to travel far from the water sources from which they emerge [33]; *H. geminatus* is no exception. Therefore, it is hypothesized that adult length is likely to be a function of the environmental conditions in the watershed within which they occurred when captured for measurements. It is unknown whether these differences in length are present in the nymphs of *H. geminatus* as well, although it is well-documented that adult size is positively correlated with final instar body size [34]. Adult size discrepancies may be indicative of potential stressors in nymphal habitat, like warming water temperatures or direct modifications to hydromorphology potentially due to changes in land cover [35]. 

*H. geminatus* is listed by the International Union for Conservation of Nature (IUCN) as globally vulnerable, vulnerable in Florida, critically imperiled in Mississippi, and imperiled in Alabama and Georgia [36], so variations in length may be signaling stressors in its aquatic habitat that, if left unnoticed, could lead to its extirpation from some of these areas. Although vulnerable, populations are currently stable and the species’ biggest threats are habitat loss due to land-cover changes, like development and forestry [36]. Our objective was to characterize the significance and extent of size differences in the species and determine whether those differences are associated with watersheds across the species’ limited range.

## 2. Materials and Methods

### 2.1. Measurement of Museum Specimens

A total of 159 adult *H. geminatus* specimens (52 females, 107 males) preserved at the International Odonata Research Institute (IORI) within the Florida State Collection of Arthropods in Gainesville were measured by J.S.G. with digital calipers. Total length (from head to tip of cerci), hindwing length, forewing length, and metathoracic femur length were measured to the nearest hundredth of a millimeter when possible. The left wings and femora were measured; right side measurements were made if necessary due to damage to left wings/femora. Body length, wing length, and metathoracic femur length were selected because they are among the standardized measurements used in species determinations [18] (pp. 8–29). Moreover, body length and wing length are relatively quick and easy measurements to make on both live and museum specimens. Of these three measurements, body length was chosen as the metric of comparison because the original species description specifically referenced body length discrepancies. Because body length was highly correlated with wing length (Table A1), using both traits is unnecessary. Metathoracic femur length was not used in further analyses due to the difficulty in measuring this trait in live individuals. 

Total length of 159 Common Sanddragon (*Progomphus obscurus*) (38 females, 121 males) adults was also measured as a means for comparison to determine whether any patterns by watershed were consistent between species rather than a species-specific idiosyncrasy; presence of consistent patterns between species would indicate external (landscape) drivers at work. This confamilial species was chosen because it occurs across a broad geographic range that encompasses *H. geminatus’* and it occurs in the same sandy streams as *H. geminatus*, often collected alongside it. *P. obscurus* specimens from IORI were only selected from collection locations within and adjacent to the geographic range of *H. geminatus*. All specimens were primarily kept in plastic envelopes with wings folded together above the bodies so that only the lateral aspect was available for measurements (Figure 1).

*H. geminatus* specimens whose bodies were broken into three or more pieces did not have their total length recorded (*n* = 19). One specimen from Tennessee was not used in analyses due to questionable identification. A total of 139 preserved specimens (46 females, 93 males) remained for use in statistical analyses after discarding the aforementioned 20 records.

### 2.2. Selection of Field Locations and Live Specimen Measurements

In addition to the measurement of preserved museum specimens, we also visited field locations to capture and measure as many live specimens as possible. We used museum specimen locations as well as expert opinion and citizen science databases (iNaturalist, OdonataCentral, GBIF) to identify specific field localities to capture *H. geminatus* adults (Table A2). Additional sampling locations were found by examining satellite imagery and ground-truthing assumed habitat for the species where streams crossed roads in the appropriate geographic range. A novel method for measuring live adult dragonflies was implemented during this study. Adults were captured with an aerial insect net and placed into a clear plastic page protector where they “perched” motionless with wings spread, allowing easy measurements to be taken with digital calipers from the dorsal aspect (Figure 2). Although using different methods for measuring museum specimens and live adults might yield slight differences, this method balanced accuracy with the consequences of collecting a sensitive species. Measuring live adults from the lateral aspect (as in museum specimens) was attempted, but not possible because restraining them in this position for accurate measurements was extremely challenging.

Total body length and the left fore- and hindwings were measured on live *H. geminatus* specimens in this way. After the individuals were sexed, their wings were marked with unique identifiers with a Sharpie marker to avoid recapture and they were photographed and released (*n* = 27; 11 females, 16 males). Metathoracic femur lengths were not measured due to the likelihood of inflated error associated with such small measurements on live specimens. Three of the hand-caught individuals were not used in analyses due to fears that their teneral measurements may not be indicative of their sexually mature dimensions.

### 2.3. Watersheds

Land-cover changes that may affect body size in insects occur across broad spatial and temporal scales. Thus, characterizing effects at a watershed scale based on the morphology of individuals within that watershed is a way to gauge watershed changes across broad temporal and spatial gradients. Many species of odonates can be hard to find due to land access issues, abbreviated flight seasons, limited geographic range, or downright scarcity. *H. geminatus* is an uncommonly encountered species with an abbreviated flight season and limited geographic range; there are not enough records of the species to justify examination of body length within fine-scale watersheds, so medium-scale 8-digit Hydrologic Unit Code (HUC-8) “subbasins” [37] were chosen as the area in which populations’ total body lengths would be examined. Body lengths in *H. geminatus* and *P. obscurus* were examined by watershed occurrence to determine (1) whether there were differences by watershed and (2) whether both species exhibited similar patterns by watershed. There are six sizes of watershed, given by hydrologic unit codes (HUCs). These range in size and specificity from large-area “regions” (HUC-2) down to small-area “subwatersheds” (HUC-12) [37]. Although characterizing watershed effects at the finest possible resolution would be desirable in many scenarios, the need to balance statistical power (higher sample size) with watershed scale in this case necessitates the use of a “medium-scale” watershed boundary, HUC-8. The HUC-8 “subbasin” scale allows enough specificity to characterize effects at a relatively fine scale but still allows a high enough sample size for statistical analyses. Some sites (and some subbasins) did not have the sample size necessary to conduct statistical analyses on each separately; sites were grouped by subbasins and subbasins were further grouped into west vs. east of the Apalachicola River for analyses. 

### 2.4. Statistical Analyses

Total lengths of museum and field specimens were compared with a non-parametric Wilcoxon rank-sum test to determine whether these data could be pooled for analysis. There were no significant differences found (*W* = 2054.5, *p* = 0.219) so the data were pooled, giving a total sample size of 166 individuals. Correlations were analyzed between total body length, fore- and hindwing lengths, and metathoracic femur lengths using a Pearson correlation test. All analyses were performed in RStudio version 2023.06.1 (R Core Team, 2023) base package *stats* (Table A1).

Length data on either side of the Apalachicola River were analyzed for significance for both species using a non-parametric Wilcoxon rank-sum test. Lengths of specimens in watersheds immediately adjacent to the Apalachicola River were also analyzed on the basis of land cover within the watershed of occurrence. Size of individual watersheds was not used as a predictor variable (Table A3). HUC-8 watersheds were downloaded to ArcGIS Pro (Esri, Redlands, CA, USA) from the USGS National Hydrography Dataset [38]. Watershed boundaries and locations of measured adults were combined using a spatial join where they overlapped. Location data were then overlaid onto a map of HUC-8 watersheds and the watershed boundaries were extracted for use in further analyses for each species (Table 1, Figure A1).

Land-cover data were downloaded from Florida Fish and Wildlife’s and Florida Natural Areas Inventory’s Florida Cooperative Landcover Map [39] and land-cover types were classified into one of six mutually exclusive categories (Upland/Sandhill, Perennial Wetland, Seasonal Wetland, Estuarine Wetland, Urban/Developed, and Rural/Agriculture). Land-cover data were clipped to subbasin polygons in ArcGIS Pro for analyses on either side of the Apalachicola River. One subbasin (03130011) was split by the river and its land cover east of the river was merged with the eastern subbasin (03120003) whereas its land cover west of the river was merged with the western subbasin (03130012). 

Impervious surface data were downloaded from the National Land Cover Dataset’s 2021 Percent Developed Imperviousness [40] as a proxy of urban development. These data were clipped to the same subbasin polygons on either side of the Apalachicola River as were the land-cover data. Percent imperviousness was batched into four categories (0–25%, 26–50%, 51–75%, and 76–100%). 

Significance of adult total lengths as a function of watershed occurrence was then assessed using a Kruskal–Wallis test, and boxplots were made using the R package *ggplot2* [41] to show total adult lengths in each watershed as well as male and female lengths in each watershed. From there, we used a pairwise Wilcoxon rank-sum test with a BH adjustment [42] to establish significant differences in total lengths amongst pairwise comparisons (Table A4 and Table A5). To understand if all *Hylogomphus* species show high levels of morphological variation, we compared the published minimum and maximum total lengths of all 6 members of the genus [43] (all occur in eastern North America) as well as the minimum and maximum lengths of individuals from our own measurements. The same was done for *P. obscurus*. 

## 3. Results

### 3.1. Patterns in Body Length

*Hylogomphus geminatus* adult lengths were originally grouped into two categories: those east of the Apalachicola River (*n* = 22) and those west of it (*n* = 144), and significant differences were found. Of the 22 specimens east of the river, 18 were museum specimens and 4 were caught and measured alive. Of the 144 specimens west of the river, 121 were museum specimens and the remaining 23 were caught and measured alive. *Progomphus obscurus* adults also showed significant differences when eastern specimens (*n* = 91) were compared to western (*n* = 68) (Figure 3). Eastern *H. geminatus* adults averaged 7.99% larger than their western counterparts; eastern *P. obscurus* averaged 2.67% larger than their western counterparts. 

When adult *H. geminatus* lengths on either side of the Apalachicola River were parsed on the basis of sex, they were significant for all comparisons: west males (*n* = 93) against west females (*n* = 51), east males (*n* = 16) against east females (*n* = 6), west males against east males, and west females against east females. *P. obscurus* showed no significance for west males (*n* = 54) against west females (*n* = 14), east males (*n* = 67) against east females (*n* = 24), and west females against east females, but did show significance for west males against east males (Figure 4). It should be noted that not all odonate species show sexual size dimorphism and, even when they do, the larger sex is species-specific [44].

### 3.2. Lengths by Watershed

After total lengths for both species were assessed on either side of the Apalachicola River, we then plotted average total lengths for each species within each subbasin of occurrence and did the same broken down by sex (Figure 5 and Figure 6). Museum and live specimen lengths for *H. geminatus* were lumped for analyses. The average length of *H. geminatus* females was significantly larger than that of males in six of the seven subbasins with enough individuals of each sex to run analyses. The average length of *P. obscurus* males and females did not significantly differ in any of the five subbasins with enough individuals of both sexes to run analyses, and females did not always average larger in each subbasin as they did in *H. geminatus*. *H. geminatus* showed significance of total length as a function of subbasin occurrence (Kruskal–Wallis: *Χ*^2^ = 77.796, *df* = 13, *p* < 0.001); there were significant differences in total lengths of 16 of the possible 91 pairwise subbasin comparisons (17.58%) (Table A4). *P. obscurus* also exhibited significant differences in total length as a function of subbasin occurrence (Kruskal–Wallis: *Χ*^2^ = 47.898, *df* = 16, *p* < 0.001) but only showed significant differences in total lengths of 5 of the possible 136 pairwise subbasin comparisons (3.68%) (Table A5).

There were eight overlapping subbasins that contained specimens of both species (Figure 7). Both species tended to show the same trends in length relative to their respective averages in six of the eight overlapping subbasins. This was most notable around the Apalachicola River. 

### 3.3. Comparisons of Body Length with Published Values

A comparison of published total lengths of the six species of *Hylogomphus* showed that *H. geminatus* is known to have the most variation in total body length (20.5% difference) followed closely by *H. parvidens* (18%) from which *H. geminatus* was split during its original species description. Three other members of the genus showed variation in body length by 5.7% or less. Our own measurements of *H. geminatus* specimens increases the published range of variation from 20.5% to 30.5%. Published values for *P. obscurus* show that the total body length of individuals varies by 12.8% but our own measurements increase that to 28% (Table A6).

### 3.4. Land-Cover Analysis

The land cover of the three subbasins surrounding the Apalachicola River (03130012, 03120003, and 03130011) was analyzed on the basis of the side of the river on which the land occurs (Table 2). The subbasins directly to the west of the river were characterized by high Rural/Agricultural influences that represented more than half of the land area. Additionally, Urban/Developed was 0.3% more common (as a function of total land area) in the west than the east. The east side of the river had proportionately more wetlands and Upland/Sandhill land cover. 

Impervious surfaces in subbasins immediately adjacent to the Apalachicola River showed negligible differences and so were not used in further analyses (Table 3). 

## 4. Discussion

### 4.1. Patterns in Length

These findings support the discrepancies noted in the original species description of *H. geminatus*, and show that both *H. geminatus* and *P. obscurus* are significantly different lengths on either side of the Apalachicola River, although the discrepancy in the former species is much greater than that in the latter. Body size in odonates may be positively associated with breeding rate and fitness components [26,28,45] (although this is not a universal rule [46]), so adults in subbasins containing larger than average individuals may not have much reason to move to other subbasins. Dispersal from breeding sites seems to be undertaken by smaller, less dominant individuals when larger, more dominant individuals are also present [45], but it is also impacted by the age and sex of individuals [47]. It is possible that whatever stressors are acting on *H. geminatus* and *P. obscurus* are affecting them to different extents, and, while the Apalachicola River does not pose a significant barrier to gene flow for either species, they may simply have no reason to fly across if they have success breeding and laying eggs around the streams from which they emerged. The differentiation could have arisen by phenotypic plasticity, but, if individuals are not crossing the river and sharing genes, this could lead to genotypic differences. If smaller individuals west of the river are able to cross it, they may not have success breeding in the presence of larger, more dominant individuals in the east, so the population in the east would still contain mostly larger individuals’ genes. This is a pattern that has been noted in the literature since the late 1970s [19]. DNA would need to be collected to assess the impact genotypic differences may have on size. Continuous stream monitoring with data loggers is another option to collect data with which we can form models to predict size but it is still difficult to assign an individual dragonfly to the stream where it was caught, as it may have developed as a nymph in an adjacent watershed. 

*H. geminatus* females were significantly larger than their male counterparts but that was rarely the case with *P. obscurus*. Although *P. obscurus* did show significant differences in length on either side of the river when sexes were grouped together, the only significant difference when lengths were parsed by sex was between east males and west males. This could suggest that eastern males have taken advantage of their larger length as a byproduct of phenotypic plasticity and have excluded smaller males from the breeding pool. Even if direct male–male competition does not play a role in the breeding strategy of *H. geminatus*, small body length in *H. geminatus* may reduce their ability to control territory in interspecific combat, and smaller males have a reduced ability to store energy, and therefore a reduced longevity and dispersal capability to search for mates [28]. However, optimal size may be those more intermediate in length as stabilizing selection would balance the benefits of large and small body length [46]. Larger females may have developed in this species to maximize egg-laying, as larger females are generally associated with higher fecundity [48,49,50] though some studies have shown no relationship between the two [51,52]. Maximizing egg-laying potential would be advantageous in *H. geminatus* because of its abbreviated flight season (roughly three months) and its historical 2:1 male-biased sex ratio evidenced by museum collection data and modern sight records. Overall, however, *P. obscurus*’ smaller effect size on either side of the river suggests that *P. obscurus* morphology may not be as closely tied to watersheds as is *H. geminatus*’ morphology. 

Relatively few *H. geminatus* individuals were found east of the Apalachicola River, both historically (museum records) and currently (field surveys). It is unknown why there was and is such a sample size discrepancy, but the area east of the river is at the eastern edge of the species’ geographic range and it is evidently unrelated to impervious surface cover. Future research, particularly focused east of the river, would increase sample sizes (and, thus, statistical power). 

### 4.2. Lengths by Watershed

Examining the significance of differences amongst pairwise subbasin comparisons (Table A4 and Table A5) supports the supposition that *P. obscurus* morphology is less closely tied to their respective watershed of occurrence than *H. geminatus* morphology. *P. obscurus* rarely showed significant differences in length amongst subbasins and did not show any significant differences between the two specific subbasins on either side of the river. *H. geminatus* had a much higher number of significant comparisons and specifically showed a large difference on either side of the river. *H. geminatus* also showed significant differences in several adjacent subbasins, which supports the cause of morphological differences being from phenotypic plasticity rather than genotype, as adjacent watersheds not separated by a river should still see some amount of gene flow. If adjacent watersheds are contributing to phenotypic differences in species, the streams within these watersheds should be monitored for potential stressors. In addition to physical factors (substrate type, stream temperature, velocity, depth, etc.) chemical assays can also be used to determine what, if any, agricultural and urban inputs are contributing to the streams and chemical factors, like pH and DO, can be compared amongst watersheds. 

When lengths of both species were analyzed in overlapping subbasins with respect to each species’ average length (Figure 7), trends within each subbasin relative to the average lengths were roughly the same for both species. Despite this, *H. geminatus* showed more significant differences across those subbasins than did *P. obscurus.* Ranges in measured lengths of both species do not suggest that *H. geminatus* has more natural variability in length than *P. obscurus* (Table A6), but *H. geminatus* does seem to exhibit larger body length differences amongst watersheds than does *P. obscurus*. This is perhaps due to the univoltine nature of *P. obscurus* at southern latitudes where it overlaps with *H. geminatus* [53]. *P. obscurus* spends less time developing in the aquatic systems it shares with *H. geminatus* as nymphs, emerging as an adult just one year after eggs are laid in the water as compared to the two years it takes *H. geminatus* to emerge from the water [18] (p. 219). *H. geminatus*’s increased development time in aquatic habitats keeps the species in contact with the stressors acting on it for longer periods of time and could explain the differences in body lengths amongst watersheds between the two species. Body lengths of other semivoltine odonates (specifically, those in the same family as *H. geminatus*) in these watersheds should be analyzed and results compared with the patterns seen in *H. geminatus* to determine whether this is true. Development and agriculture can have significant effects on the voltinism of aquatic macroinvertebrate assemblages; as natural landscapes are converted more to urban and agricultural areas, macroinvertebrates that spend more time in the water may become extirpated, leaving mostly univoltine species in their place [54]. The same stressors that can lead to the eventual extirpation of semivoltine species may be causing significant plasticity in *H. geminatus* body lengths and could eventually lead to their extirpation from highly urbanized or agriculturalized areas. 

### 4.3. Comparison of Length with Published Values

A summary of published lengths of the species within *Hylogomphus* shows it is not a highly morphologically variable genus but the species from which *H. geminatus* was split does exhibit similar morphological variation based on published values (Table A6). Our measurements also showed higher variability in total lengths for both *H. geminatus* and *P. obscurus* than published values showed. Intraspecific size variability of odonates is not a new concept [55,56], but has typically been attributed to differences in climate and water temperature based on latitude, although they can also vary within latitude [17]. Climate, including temperature, should not have an effect on *H. geminatus*, whose geographic range is only across a small area on a mostly longitudinal gradient. Primary productivity has been suggested as a mechanism predicting odonate body size [57] and should be investigated as a factor in the streams inhabited by *H. geminatus*. 

### 4.4. Land-Cover Effects on Length

The differences in body length that were observed may have been due to changes in land cover from urban development, deforestation, streambank destabilization, and stream channelization [17]. In the United States, the state of Florida has experienced relatively recent and rapid land-cover changes resulting from increases in the extent of urbanization and agriculture since the early 1900s [58,59]. Florida increased in population by 2.95% per year between 1960 and 1997 [60], associated with a 60% increase in agricultural lands between 1936 and 1995, and a 632% increase in urbanized lands during the same time period [58]. 

Thinner canopies via deforestation can lead to more sunlight permeation in streams, which is associated with warming water. Warmer water temperatures may be correlated with decreased body size in odonates [61,62], but the relationship between temperature and body size is complex, with some studies finding no effect on body size at emergence [10,62] or even a positive relationship [63]. There also may be indirect effects on odonates, such as impacts on attack rate and food handling time [64,65,66], which are crucial functional responses for an organism’s metabolic rate and, therefore, its developmental rate. Furthermore, channelization of naturally dendritic streams causes a reduction in spatial heterogeneity and increases in stream depth and velocity, which not only affects odonates but also the prey on which they depend [67,68].

Warming waters also have a reduced ability to hold DO, a stressor that has been shown to reduce overall odonate diversity in streams [69] and could trigger changes in ecomorphologically variable taxa. Streambank destabilization can exacerbate erosion, causing increases in turbidity and sedimentation which can reduce an organism’s ability to successfully find food, limit light permeation necessary for aquatic primary producers, and clog the delicate gills of DO-breathing organisms, and synergistic effects amongst the aforementioned consequences of streambank destabilization can have further negative effects on aquatic biota [70,71]. Agricultural and urban runoff pollute streams with heavy metals, salts, and other chemicals that change stream pH, total dissolved solids, and specific conductance, reducing water quality for aquatic macroinvertebrates. Water pollution is rarely limited to a single stretch of stream at the source of the disturbance, but generally travels much further downstream, affecting the large majority of any watershed within which the disturbance occurs. First-order headwater streams in particular supply roughly 55% of the mean-annual water volume received by higher-order navigable rivers [72], so even small-scale, point-source pollution in low-order streams has the potential to affect a large extent of the landscape and therefore impact a great deal more aquatic taxa than just those immediately around the disturbance.

Those pollutes may have been in the Apalachicola River drainage since the original species description of *H. geminatus*. Populations of *H. geminatus* directly west of the Apalachicola River have consisted of smaller individuals [19] which is especially concerning given that more than half of the area within the subbasins to the immediate west of the river is comprised of agricultural land. Not only does agricultural development reduce the number of viable streams in which *H. geminatus* can live through direct waterway modification, it also reduces the amount of upland/sandhill habitat that deposits the sediments into the streams on which *H. geminatus* is so dependent. *Hylogomphus geminatus* is only found in streams with primarily sandy substrate and, as that substrate disappears due to agricultural development, more silt may be deposited into the streams instead, creating a possible stressor that negatively impacts benthic macroinvertebrates [73]. The smaller individuals in these areas could be signaling decreased fitness due to environmental stressors [28,45,46] that may eventually lead to extirpation of this and similar species. Agricultural practices are commonly associated with high levels of nitrogen (N) and phosphorus (P) runoff [74] as well as various herbicides and pesticides [75]. This is especially concerning in waterways because aquatic organisms can bioaccumulate these chemicals and may be more susceptible to them than terrestrial organisms [75], and while collateral damage to non-target organisms in agricultural settings may be limited, once chemicals enter nearby streams, side effects can be drastic. In one study, heavy metals in a stream in Malaysia were shown to have bioaccumulated to levels in damselfly tissue beyond that of levels in stream sediments [76]. Furthermore, chemicals are carried further away by streams, often ending up as far as the ocean, and many more non-target organisms fall victim to them [77]. 

However, *H. geminatus* east of the Apalachicola River are known to be larger than their western conspecifics. Rural/Agricultural land cover east of the river is still relatively high (proportionately just over a third of total land area) but much more of the land area is still covered by native wetlands and adjacent upland/sandhill habitat that deposits the vital substrate to which *H. geminatus* is so closely tied. *Hylogomphus geminatus* total length is significant as a function of the watershed in which they occur. These trends should be verified with more focused land-cover analyses, especially with respect to the temporal scale, and an increase in N, P, and pesticide monitoring in agricultural-adjacent waterways. 

Understanding the real-time changes in biotic and abiotic variables within the specific waterways in which *H. geminatus* exists would require continuous stream monitoring with permanent data loggers. Significant differences amongst adjacent watersheds serve as an early warning sign that some aquatic aberrations are inducing effects in the aquatic community; those effects may directly impact *H. geminatus*’ size or indirectly by decreasing available prey. Continuous stream monitoring can submit real-time data, but dragonflies can live in streams for up to five years as aquatic nymphs [78,79,80]. Their morphology and/or presence may thus be indicative of environmental aberrations across a large temporal scale. These effects suggest not only a threat to resources on which we depend, but also on the existence of rare and endemic species as well as those of economic importance within the core range of *H. geminatus*. Future research should include DNA sampling within different watersheds but especially on either side of the Apalachicola River, in order to rule out genotypic differences. Data loggers should be implemented in stretches of streams where *H. geminatus* is known to occur to monitor stream variables and build more mechanistic predictive models. Quantitative aquatic macroinvertebrate surveys should be used to assess the density and diversity of prey species available to *H. geminatus* nymphs. Satellite imagery should be used to assess the effects of terrestrial changes (like land clearing, urbanization, and agriculture) that might impact adjacent waterways. Finally, quantitative morphological analyses should be used to assess differences among the *H. geminatus* core population, disjunct populations, and *H. parvidens* (e.g., patterns and colors on individuals could be quantified, and the morphology of mating appendages could be measured in detail for comparisons). Analyses like these would allow us to assess the scale of morphologic differences. In general, more data are needed to assess what the significance of these differences is, and more specimens from areas east of the river where *H. geminatus* seems to be less common should be measured and analyzed.

As the subbasins around the core range of *H. geminatus* continue to transform due to anthropogenic activities, this uncommon species could see retractions in its distribution and eventual extirpation from areas where its optimal habitat no longer exists. Understanding the causes of variation in its ecomorphology now may allow us to prevent future losses of this species and many others that depend on its crucial freshwater systems. 

## Figures and Tables

**Figure 1 insects-14-00754-f001:**
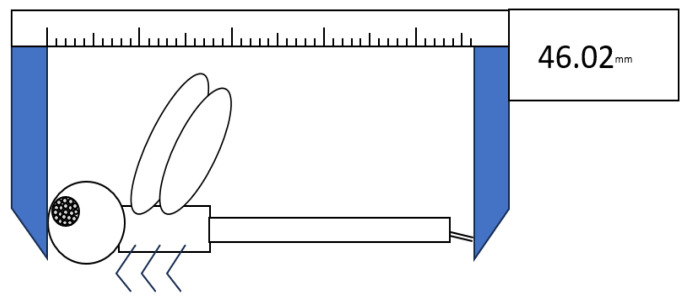
The general approach to measuring total length of odonate specimens (from head to tip of cerci) housed in plastic envelopes at IORI.

**Figure 2 insects-14-00754-f002:**
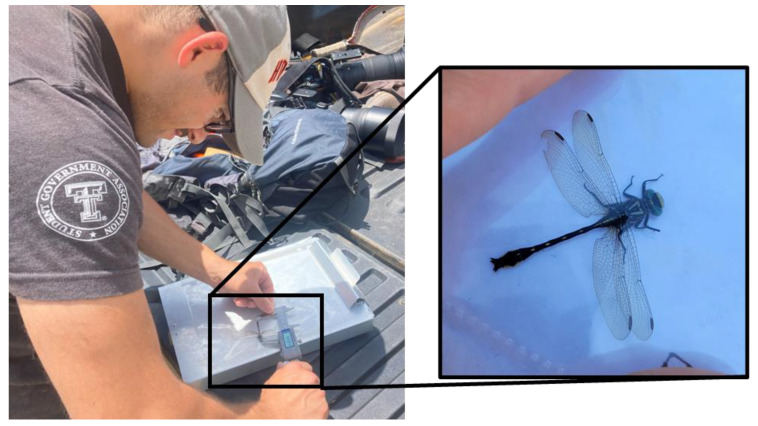
The general approach to measuring total length of live odonate specimens by temporarily placing them in a clear page protector and measuring head to tip of cerci across the dorsal aspect.

**Figure 3 insects-14-00754-f003:**
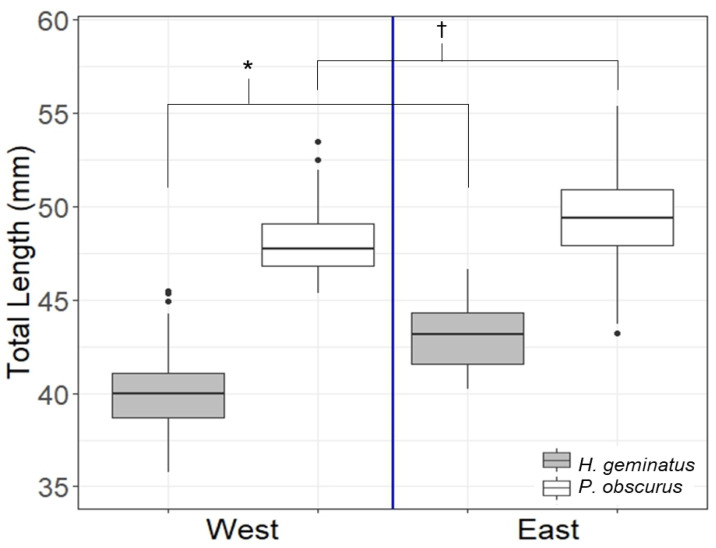
*H. geminatus* individuals averaged significantly larger east of the Apalachicola River (represented by the vertical blue line) than west (*W* = 2793, *p* < 0.001). *P. obscurus* individuals also showed the same significant trend (*W* = 4246, *p* < 0.001). Significance between *H. geminatus* comparisons is denoted with an asterisk and that of *P. obscurus* is denoted by a cross.

**Figure 4 insects-14-00754-f004:**
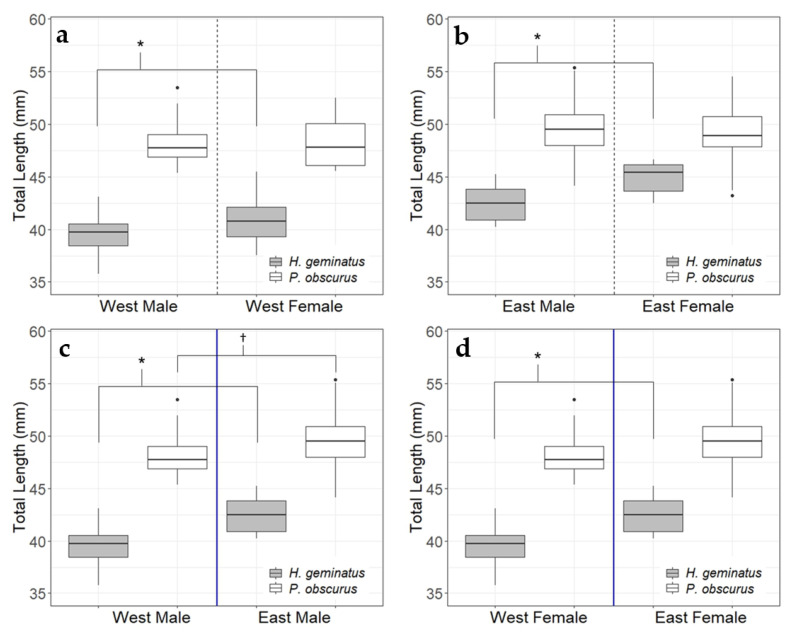
All comparisons between *H. geminatus* individuals of either sex on either side of the Apalachicola River showed significant differences whereas *P. obscurus* only showed significant differences in length between east and west males. Significance between *H. geminatus* comparisons is denoted with an asterisk and that of *P. obscurus* is denoted by a cross. Note that the vertical blue line separating western and eastern data in (**c**,**d**) represents the Apalachicola River. (**a**) Western female *H. geminatus* were significantly larger than western males (*W* = 1480, *p* < 0.001) but there was no significant difference for *P. obscurus* (*W* = 379, *p* = 0.497); (**b**) eastern female *H. geminatus* were significantly larger than eastern males (*W* = 15, *p* = 0.007) but there was no significant difference for *P. obscurus* (*W* = 880.5, *p* = 0.247); (**c**) eastern male *H. geminatus* and *P. obscurus* were significantly larger than their western conspecifics (*W =* 135.5, *p* < 0.001) and (*W* = 1054, *p* < 0.001), respectively; (**d**) eastern female *H. geminatus* were significantly larger than western females (*W* = 21, *p* < 0.001) but there was no significant difference for *P. obscurus* (*W* = 130, *p* = 0.130).

**Figure 5 insects-14-00754-f005:**
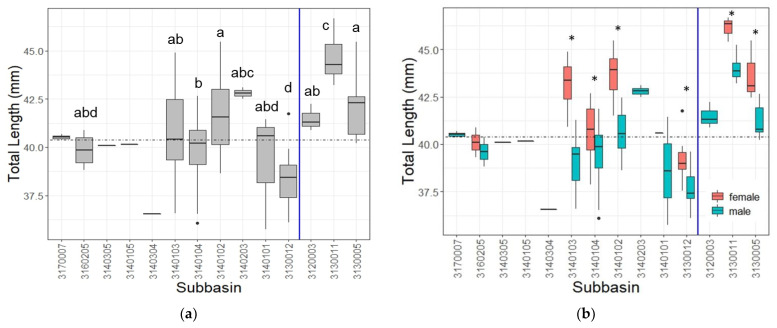
*H. geminatus* total lengths within each subbasin (simplified HUC-8 codes from Table 1 without the leading 0). Note that the vertical blue lines separating subbasins in (**a**,**b**) represent the Apalachicola River and its placement relative to associated subbasins. The horizontal dashed line represents the average total length of all *H. geminatus* specimens measured. (**a**) *H. geminatus* varied in length significantly in 16 pairwise subbasin comparisons (Table A4). This is especially noticeable in subbasins directly on either side of the river. Significant comparisons are denoted by letters above the box plots. HUCs denoted with the same letters are not significantly different. Those without any letters have no significant comparisons. (**b**) *H. geminatus* total lengths by subbasin parsed by sex. Females averaged larger than males for all subbasins. Significant differences between male and female lengths are seen in 6 of the 7 subbasins with enough specimens of each sex to assess significance. Subbasins containing significant differences between males and females are denoted with an asterisk.

**Figure 6 insects-14-00754-f006:**
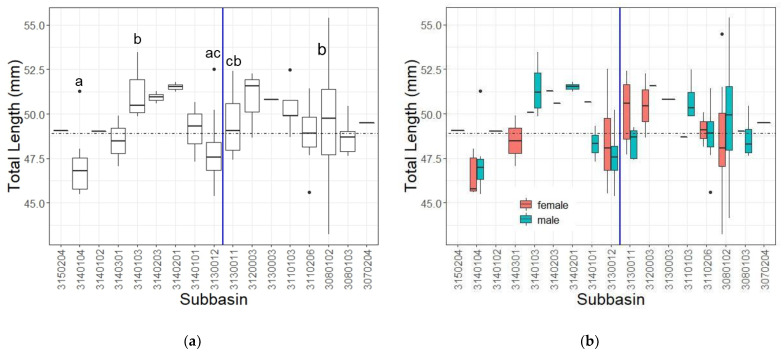
*P. obscurus* total lengths within each subbasin (simplified HUC-8 codes from Table 1 without the leading 0). Note that the vertical blue lines separating subbasins in (**a**,**b**) represent the Apalachicola River and its placement relative to associated subbasins. The horizontal dashed line represents the average total length of all *P. obscurus* specimens measured. (**a**) Analyses of total length in each subbasin revealed significant differences in 5 pairwise comparisons (Table A5). Significant comparisons are denoted by letters above the box plots. HUCs denoted with the same letters are not significantly different. Those without any letters have no significant comparisons. (**b**) *P. obscurus* total lengths by subbasin parsed by sex. Females were not always larger than males and no significant differences between male and female lengths were found in any subbasin with enough of each sex to assess significance.

**Figure 7 insects-14-00754-f007:**
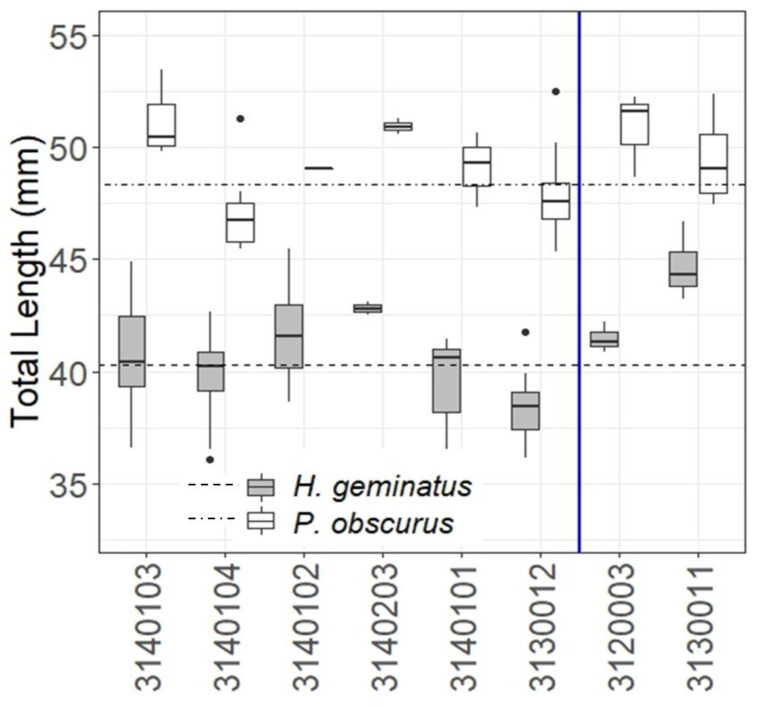
Average lengths of *H. geminatus* and *P. obscurus* in 8 overlapping subbasins (simplified HUC-8 codes from Table 1 without the leading 0). The vertical blue line separating subbasins represents the Apalachicola River and its placement relative to associated subbasins. The dashed horizontal lines represent the overall average lengths of the respective species. Notably, the subbasin immediately west of the river (03130012) contained smaller individuals of both species and the one immediately east of the river (03120003) contained larger individuals of both species.

**Table 1 insects-14-00754-t001:** HUC-8 watersheds where *H. geminatus* and *P. obscurus* occurred and the associated sample size in each watershed. *P. obscurus* specimens were collected from a wider geographic range than *H. geminatus*, hence the additional number of codes.

*Hylogomphus geminatus*	*Progomphus obscurus*
03130011 (*n* = 10)	03150204 (*n* = 1)
03140105 (*n* = 1)	03140104 (*n* = 13)
03140104 (*n* = 69)	03140102 (*n* = 1)
03130012 (*n* = 29)	03140301 (*n* = 2)
03140103 (*n* = 10)	03140103 (*n* = 5)
03130005 (*n* = 9)	03140203 (*n* = 2)
03140304 (*n* = 1)	03140201 (*n* = 2)
03140305 (*n* = 1)	03140101 (*n* = 3)
03140102 (*n* = 22)	03130012 (*n* = 39)
03170007 (*n* = 2)	03130011 (*n* = 11)
03140101 (*n* = 3)	03120003 (*n* = 3)
03140203 (*n* = 2)	03130003 (*n* = 1)
03160205 (*n* = 4)	03110103 (*n* = 5)
03120003 (*n* = 3)	03110206 (*n* = 13)
	03080102 (*n* = 52)
	03080103 (*n* = 5)
	03070204 (*n* = 1)

**Table 2 insects-14-00754-t002:** Florida land-cover data for three watersheds surrounding the Apalachicola River. Both sides had relatively high Rural/Agricultural inputs, but the west side of the river had notably more and was also characterized by less Upland/Sandhill habitat and overall wetlands.

Land-Cover Type	West of River	East of River
Upland/Sandhill	10.9%	21.7%
Perennial Wetland	20.8%	25.6%
Seasonal Wetland	8.5%	12.7%
Estuarine Wetland	0.3%	1.3%
Urban/Developed	5.5%	5.2%
Rural/Agriculture	54.1%	33.5%

**Table 3 insects-14-00754-t003:** Percent of landscape denoted as impervious land cover in the HUC-8 subbasins on either side of the Apalachicola River, Florida.

Impervious Coverage (%)	West of River	East of River
0–25	98.40%	98.59%
26–50	0.99%	0.90%
51–75	0.46%	0.38%
76–100	0.15%	0.14%

## Data Availability

Data available in a publicly accessible repository. The data presented in this study are openly available in GitHub at https://github.com/Joseph73192/H_geminatus_adults (accessed on 6 September 2023).

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
