# Peer review of "Watershed-Mediated Ecomorphological Variation: A Case Study with the Twin-Striped Clubtail Dragonfly (Hylogomphus geminatus)"

_insects, 2023, doi:10.3390/insects14090754_

Round 1

Reviewer 1 Report

Dear authors, I congratulate you on the study, without a doubt a study that will be widely read and cited. I leave some suggestions to improve the reader's understanding of the manuscript.

  I had a lot of difficulty reading the manuscript, I found it complex. I missed a justification of how the size of the basin could affect the size of Odonata, the impact I think it was well described.

In the material and methods section I read as if I were going to replicate your work and I found it difficult to understand how you chose the collection points, how many individuals you sampled at these points, how you measured the basin, how many points within each basin were sampled, how many measured individuals of each species, male in female in each basin. How did you control for the effect of environmental change when you went to test the effect of basin size on males and females of both species. and the same when they went to test the effect of environmental change. I didn't understand how it later unfolded into two areas in the results. I didn't understand the test between males and females, in general isn't the female always bigger?

Excuse me if these questions are very banal, but they were doubts that prevented me from understanding your analyzes and results.

Best regards

Reviewer 2 Report

Please see the comments in the file.

Round 2

Reviewer 1 Report

No comments.

Author Response

Thank you for your comments on the first draft of the manuscript and thank you again for your ratings on the second draft. 

Reviewer 2 Report

The authors significantly impreved the qualit of the manuscript and I am happy with the current version. I have only one minor comment: in line 385 [and perhaps other places in the text] change "fitness" to "fitness components" or  remove ref. 26 as this paper did not test for true fitness, but only traits linked to fitness. I wish you success in the further stages of manuscript submission!

Author Response

Thank you for your comments on both versions of our manuscript, we have changed "fitness" to "fitness components" on line 385, but left one other use of the word, "fitness," as it seems more appropriate in its context. Thank you, again for your kind words. 

Sincerely, Joe Girgente and Nancy McIntyre